# Application of Humanized Zebrafish Model in the Suppression of SARS-CoV-2 Spike Protein Induced Pathology by Tri-Herbal Medicine Coronil via Cytokine Modulation

**DOI:** 10.3390/molecules25215091

**Published:** 2020-11-02

**Authors:** Acharya Balkrishna, Siva Kumar Solleti, Sudeep Verma, Anurag Varshney

**Affiliations:** 1Drug Discovery and Development Division, Patanjali Research Institute, NH-58, Haridwar 249 405, Uttarakhand, India; pyp@divyayoga.com (A.B.); siva.kumar@prft.co.in (S.K.S.); sudeepverma@prft.co.in (S.V.); 2Department of Allied and Applied Sciences, University of Patanjali, Patanjali Yog Peeth, Roorkee-Haridwar Road, Haridwar 249 405, Uttarakhand, India

**Keywords:** humanized zebrafish, SARS-CoV-2 infection, anti-inflammatory, behavioral fever, swim bladder, Coronil, herbal medicine, Ayurveda

## Abstract

Zebrafish has been a reliable model system for studying human viral pathologies. SARS-CoV-2 viral infection has become a global chaos, affecting millions of people. There is an urgent need to contain the pandemic and develop reliable therapies. We report the use of a humanized zebrafish model, xeno-transplanted with human lung epithelial cells, A549, for studying the protective effects of a tri-herbal medicine Coronil. At human relevant doses of 12 and 58 µg/kg, Coronil inhibited SARS-CoV-2 spike protein, induced humanized zebrafish mortality, and rescued from behavioral fever. Morphological and cellular abnormalities along with granulocyte and macrophage accumulation in the swim bladder were restored to normal. Skin hemorrhage, renal cell degeneration, and necrosis were also significantly attenuated by Coronil treatment. Ultra-high-performance liquid chromatography (UHPLC) analysis identified ursolic acid, betulinic acid, withanone, withaferine A, withanoside IV–V, cordifolioside A, magnoflorine, rosmarinic acid, and palmatine as phyto-metabolites present in Coronil. In A549 cells, Coronil attenuated the IL-1β induced IL-6 and TNF-α cytokine secretions, and decreased TNF-α induced NF-κB/AP-1 transcriptional activity. Taken together, we show the disease modifying immunomodulatory properties of Coronil, at human equivalent doses, in rescuing the pathological features induced by the SARS-CoV-2 spike protein, suggesting its potential use in SARS-CoV-2 infectivity.

## 1. Introduction

Coronavirus disease 2019 (COVID-19) is a severe respiratory tract infection caused by the newly emerged coronavirus named, novel severe acute respiratory syndrome coronavirus-2 (SARS-CoV-2). This has posed a serious global public-health emergency [1] and pushed the world into chaos. More than 40 million people have been infected and the death toll is rapidly increasing, as reported by the World Health Organization (WHO) (https://covid19.who.int/) [2]. Despite global efforts to contain COVID-19, the pandemic is rapidly spreading, calling for immediate need to identify the clinically-relevant prophylaxis and therapeutic strategies.

Coronaviruses are enveloped large plus-strand ribonucleic acid (RNA) viruses that belong to the Coronaviridae family [3]. In 2002, SARS-CoV, a coronavirus, was identified as the causative agent of severe acute respiratory syndrome (SARS) [4]. Another coronavirus called Middle East respiratory syndrome-CoV (MERS-CoV) also causes highly pathogenic respiratory tract infections [5]. The novel coronavirus causing the COVID-19 pandemic is a beta corona virus and shares phylogenetic similarities to both SARS-CoV and MERS-CoV and was named severe acute respiratory syndrome coronavirus 2 (SARS-CoV-2) [6,7].

Interaction between the SARS-CoV homo-trimeric spike (S) glycoprotein on the envelope and the host cell receptor angiotensin-converting enzyme 2 (ACE2) triggers a cascade of events leading to the fusion of the viral lipid bilayer to the host cell membrane, leading to viral entry [8,9]. SARS-CoV primarily targets cells of the airway epithelium, alveolar epithelium, vascular endothelium, and lung macrophages, all of which express the ACE2 receptor [10] and are also targeted by SARS-CoV-2 [11]. Interestingly, viral infection reduces ACE2 expression in the lungs and it may be a key feature for its pathology [12]. ACE2 regulates the renin-angiotensin system (RAS), which is important for regulating blood pressure and fluid/electrolyte balance, and reduced levels during infection can enhance inflammation and vascular permeability in the airways.

SARS-CoV-2 infection of the lungs results in accumulation of macrophages and monocytes, and releases cytokine like IL-1β, TNF-α, IL-6, and IFN-γ to initiate adaptive T and B cell immune responses for the resolution of infection [7]. A dysfunctional immune response can result in a cytokine storm and cause severe lung and systemic pathology [13]. This can also be initiated by antagonism of interferon response by SARS-CoV viral proteins, leading to excessive inflammation [14]. Furthermore, elevated cytokine levels cause multi organ failure and myocardial damage [15]. Geriatric population and people with co-morbidities are more prone to infection due to compromised immune responses. However, important host immune factors responsible for the disease severity and inflammatory responses in some patients remain poorly defined.

Use of various in vitro and in vivo models to study and understand SARS-CoV-2 pathology using cell lines, organoids as well as small and large animal models is in progress [16]. Reliable animal models that can reflect the COVID-19 clinical symptoms are being investigated and developed, in order to decipher the pathophysiology of the SARS-CoV-2 infection [17]. While primate and non-primate models have been the choice of experimental animal models, applicability of the zebrafish model for the screening of novel therapeutics and modulators of biological processes in vivo cannot be over looked. In the case of SARS-CoV-2, this has become even more relevant as the virus does not readily infect the murine and canine hosts, and creating a humanized murine model is rather resource and time intensive.

Zebrafish is a small and versatile organism that is very easy to handle and the signaling mechanisms, interaction with chemical modifiers, and the host–virus communications on mucosal tissues are amenable to analysis with delicate detail [17]. Notwithstanding obvious physiological and biological differences, zebrafish has been a model of choice for studying human diseases including virus-induced diseases [18,19] and to discover new drugs or to repurpose existing drugs [20]. Zebrafish has a well-defined innate and adaptive immune systems and shares a remarkable similarity to human counterparts [21]. Furthermore, the interaction between viral glycoproteins and olfactory sensory neurons of the fish nasal region are well understood [22]. Several human viruses including chikungunya and influenza can colonize zebrafish [23], making it a very attractive and alternative model system. Unlike murine models, zebrafish have swim bladders as buoyancy organs, and human cells could be introduced to swim bladders for the generation of xeno-transplanted humanized models for respiratory diseases like SARS-CoV-2 infection. This approach has been successfully used for modeling lung cancer in zebrafish [24]; and in modeling Chronic Obstructive Pulmonary Disease (COPD) [25] and *P. aeruginosa* pathogenesis [26].

Currently, various therapies to target the immune function of SARS-CoV-2 infection are in various phases of clinical trials. Use of dexamethasone (DEX) has been associated with reduced death and improved recovery of patients receiving mechanical ventilation or oxygen support [27]. In additional to modern small molecule compounds, the use of traditional Chinese medicine such as the poly-herbal formulation Lianhuaqingwen for SARS-CoV-2 pathology treatment has shown some benefits as an anti-viral and anti-inflammatory agent [28,29]. Therefore, exploring herbal medicines [30] in the present pandemic situation could be highly advantageous.

Coronil is a tri-herbal medicine developed in India, specifically as a remedy for SARS-CoV-2. Coronil consists of enriched extracts of *Withania somnifera* (L.) Dunal (Ashwagandha), *Tinospora cordifolia* (Willd.) Miers (Giloy, heart-leaved moonseed), and *Ocimum sanctum* L (Tulsi, holy basil) plants. The immunomodulatory [31,32,33] and anti-viral properties of these plants and their phytochemical components have been studied using in silico, in vitro, and in vivo approaches [34].

The aim of the present study was to profile the chemical composition of Coronil using ultra-high-performance liquid chromatography (UHPLC) and to evaluate the beneficial effects of Coronil against SARS-CoV-2 spike protein induced pathological symptoms in a humanized zebrafish model. The glucocorticoid drug, dexamethasone (DEX), was tested concurrently in studying the clinically relevant parameters. In addition, we validated the anti-inflammatory properties of Coronil using human alveolar epithelial cells, A549 for the IL-6 and TNF-α cytokine response as well as a reporter cell line for NF-κB transcriptional activity. Taken together, our results indicate the potential beneficiary effects of Coronil against SARS-CoV-2 related pathology.

## 2. Results

### 2.1. Ultra-High-Performance Liquid Chromatography (UHPLC) Analysis of Coronil Detected the Presence of Bio-Active Metabolites

Coronil consists of the whole plant extract of *W. somnifera*, stem extract of *T. cordifolia*, and leaf extract of *O. sanctum*. To identify the bio-active ingredients present in the Coronil, UHPLC analysis was performed. Comparing the chromatograms obtained from the reference standards, UHPLC identified the presence of ten signature molecules (Figure 1) namely, ursolic acid and betulinic acid (detected at 210 nm), withanone, withaferine A, withanoside IV, withanoside V, cordifolioside A and magnoflorine (detected at 227 nm) as well as rosmarinic acid and palmatine (detected at 325 nm) at varying concentrations (μg/mg), as shown in Table 1. All test samples were separated within 65 min at different retention times in a chromatography column.

### 2.2. Generation of Humanized Zebrafish (HZF) Model and Induction of SARS-CoV-2 Spike Protein Stimulated Pathology

Humanized zebrafish model was generated by xeno-transplanting human alveolar epithelial cells (A549) in the posterior lobe of the swim bladder in adult zebrafish. We then analyzed the anatomy and cytology (Figure 2a,b) after seven days, in order to confirm the xeno-transplantation and successful adherence of human respiratory epithelial cells.

The anterior and posterior lobes of the swim bladder in normal control fish (NCTL) (Figure 2a(i)) appear normal in size, shape, and color with continuous tunica externa and mesothelium lining of smooth muscle. Xeno-transplantation of A549 cells did not induce any abnormalities in the swim bladder of HZFCTL (Figure 2a(ii)) and appeared structurally similar to NCTL with normal morphological traits at the time point tested. Next, we analyzed the cytology of the swim bladder and identified the presence of spherical nucleated squamous epithelia, oval nucleated columnar epithelia, smooth muscle cells with elongated nuclei, and the presence of eosinophilic collagen in NCTL (Figure 2a(iii)). Furthermore, xeno-transplantation of A549 cells in the HZFCTL (Figure 2a(iv)) group also exhibited normal cytological features similar to NCTL at the time points tested. In addition, cyto-smears also confirmed the presence of human alveolar epithelial cells in the swim bladders of HZFCTL fish (Figure 2a(iv)).

### 2.3. Study Design for Translational Dosing

Post seven-days confirmation of xeno-transplantation of A549 cells, the fish were injected with 2.8 ng SARS-CoV-2 spike protein (DCTL) and tested for the induced pathological features (Figure 2b). We then attempted to test for the pharmacological effects of the tri-herbal medicine, Coronil. The translational dose of Coronil (CN) was determined at 1000 times less than the relative human doses (two tablets each of 640 mg, thrice per day, ~4000 mg/day), based on the differences in body weights and body surface area of adult zebrafish and humans. In the present study, we tested two different translational doses of Coronil at two different time points, in order to understand the acute and subacute pathophysiological features (Figure 2b).

Zebrafish treated with Coronil (CN) translated from the human prescribed daily dose (4000 mg/day) included 0.2X (CN12, 12 μg/kg) and 1X (CN58, 58 μg/kg) doses were screened after three days (fourth day end point) and six days (seventh day) post spike protein challenge (Figure 2b). Similarly, the dexamethasone (DEX) dose translated from the human prescribed dose (6 mg/day) was determined to be 0.08 μg/kg. At the end of the study time points, we tested for anatomy and cytology of the swim bladder and kidney for the presence of skin hemorrhage and behavioral fever.

### 2.4. Coronil Inhibits SARS-CoV-2 Spike Protein Induced Mortality of Zebrafish

In a parallel study, the survival and mortality of zebrafish in all the test groups (Figure 2c) for a longer duration was assessed. Results indicated that the SARS-CoV-2 spike protein in DCTL induced 8.7% mortality after two days and 13.8% mortality after five days. Twenty percent mortality after eight days resulted in 80% survival by the end of the study period at 10 days in DCTL group. In contrast, CN treatment or DEX treatment completely rescued from SARS-CoV-2 spike protein induced mortality, resulting in 100% survival during this study period, which was significantly higher (*p* < 0.001) to the DCTL group, and was similar to the NCTL and HZFCTL test groups.

### 2.5. Coronil Reduces SARS-CoV-2 Spike Protein Induced Behavioral Fever

Fever is one of the common symptoms in SARS-CoV-2 infection, as advised by the WHO (https://www.who.int/health-topics/coronavirus). Zebrafish, as an ectotherm, exhibits behavior fever as an adaptive immune response. We tested for SARS-CoV-2 spike protein induced behavioral fever in zebrafish by recording the time spent by fish at three temperature gradient chambers (i.e., 23 °C, 29 °C, and 37 °C).

As expected, both NCTL and HZFCTL fish spent more time at 29 °C on both the fourth day and seventh day time points and less time at 37 °C (Table 2 and Figure 3). The spike protein challenge in DCTL resulted in behavioral fever and the fish spent more time at 37 °C and lesser time at 29 °C at both the four day and seventh day time points (Figure 3). DEX admiration had moderate change in behavioral fever only on the seventh day, and did not show a change on the fourth day as they spent the majority of time at 37 °C (Figure 3). Interestingly, Coronil treatment moderately rescued from behavioral fever on the seventh day in a dose dependent manner, as indicated by a modest increase in the time spent at 29 °C compared to DCTL. This indicates the protective effect of Coronil on thermo-regulatory behavior (Table 2), which is the cardinal sign of an immunomodulatory effect.

### 2.6. Swim Bladder Analysis Identified the Protective Response of Coronil

Next, we tested for the swim bladder morphology (Figure 4). The results indicated that, while the normal CTL (NCTL) and HZFCTL appeared structurally normal (Figure 4a,b,g,h), SARS-CoV-2 spike protein inoculation in disease control (DCTL) induced inflation of the anterior lobe, suggesting structural anomaly and edema on the fourth day (Figure 4c). This further deteriorated and the anterior lobe of the swim bladder collapsed on the seventh day (Figure 4i). While DEX treatment in the fourth day group showed a mildly inflated swim bladder with intact tunica externa and gas gland (Figure 4d), in the seventh day DEX group, the anterior lobe was enlarged and the posterior lobe was narrowed due to an edematous gas gland, indicating structural anomaly, and no apparent rescue (Figure 4j). At the fourth day time point, all the fish treated with both doses of Coronil (12 μg/kg and 58 μg/kg) showed normally inflated swim bladder lobes with apparent size, shape, and color (Figure 4e,f) that were comparable to the NCTL and HZFCTL groups. Similarly, at the seventh day time point, the CN12 and CN58 groups exhibited normal morphological traits in size, shape, and color with continuous tunica externa and mesothelium lining of smooth muscle (Figure 4k,l).

### 2.7. Coronil Attenuates SARS-CoV-2 Spike Protein Induced Inflammation in Swim Bladder

We next analyzed swim bladder cyto-smears for the cellular profiles (Figure 5) and identified the presence of squamous epithelium, columnar epithelium, smooth muscle cells, and collagen in NCTL and HZFCTL (Figure 5a,b,g,h) at both the time points tested. Compared to HZFCTL, on the fourth day, the DCTL group exhibited significant infiltration of granulocytes and macrophages along with the epithelial cells and the myocytes (Figure 5c,n,o). This indicates a robust inflammatory response initiated by the presence of cellular, nuclear debris, and edema (5c). Interestingly, on the seventh day, in the DCTL group, no inflammatory cells were identified, however, cellular materials and edema were detected (5i). Compared to DCTL, in the DEX group, at the fourth day time point, the number of leukocytes and macrophages were significantly decreased and the number of lymphocytes were significantly elevated (Figure 5d,n,o,p). Whereas, at the seventh day time point, a significant but moderate increase in the number of macrophages and lymphocytes and cellular debris indicated rescue from immune response by DEX (Figure 5j,n,o,p).

Coronil treatment resulted in a dose dependent reversal of cytological disease markers at both the time points tested. Compared to DCTL, at the fourth day time point, the granulocytes and macrophages were significantly decreased in a dose dependent manner (Figure 5e,f,n,o). Interestingly, a significant increase in the lymphocyte count was noticed at CN12, whereas CN58 showed a mild lymphocyte infiltration (Figure 5p). At the seventh day time point, while no granulocytes and macrophages were noticed in both the CN treated groups (Figure 5k,l,n,o), the number of lymphocytes were significantly increased in CN12 (Figure 5p). Taken together, mild to moderate cellular degeneration, substantial reduction of inflammatory cellular debris, along with the presence of normal epithelial cells and myocytes in the CN treated groups indicated a disease rescue morphology.

### 2.8. Coronil Inhibits SARS-CoV-2 Spike Protein Induced Renal Cell Necrosis

Next, we tested for the gross anatomy of the zebrafish kidney, and identified that NCTL and HZFCTL (Figure 6a,b,g,h) exhibited normal morphological traits, with well-defined internal arrangements. Spike protein challenge in DCTL resulted in the loss of kidney tubular segmentation and induced vascular degeneration indicative of renal necrosis (Figure 6c,i). DEX treatment showed normal renal architecture at the 4th day time point whereas, the 7th day time point exhibited mild structural abnormality (Figure 6d,j). Remarkably, CN treatment displayed normal renal morphology at both doses of CN12 and CN58 (Figure 6g,f,k,l) at both time points tested.

We also examined the kidney smears for renal pathological features. NCTL and HZFCTL smears at both the fourth and seventh day time points exhibited a well arborized tubular network of mesonephros with evenly scattered glomerulocytes without any observable signs of necrosis and degeneration (Figure 7a,b,g,h). Spike protein challenge in DCTL induced disorganized cellular arrangement and erythrocyte aggregation in a time dependent manner (Figure 7c,i). Accordingly, necrotic and degenerating renal cells were also significantly increased in a time dependent manner (Figure 7c,i,m,n), confirming spike protein induced renal damage. DEX treatment significantly decreased both necrotic and degenerating cells (Figure 7d,j,m,n), suggesting rescue in renal pathology. Importantly, Coronil treatment rescued from renal cell degeneration and necrosis. CN treatment at the fourth day time point rescued spike protein induced renal necrosis in a dose dependent fashion at both doses (CN12 and CN58). Observed recovery in degeneration, along with disorganized renal epithelial arrangements (Figure 7e,f,m,n), indicated a gradual improvement from thee DCTL phenotype. Similarly, at the seventh day time point, a significant dose dependent change in the % necrosis was accompanied by significant decrease in the % renal degradation (Figure 7k,l,m), indicating a dose dependent gradual recovery of the renal cytology profile.

### 2.9. Coronil Attenuates SARS-CoV-2 Spike Protein Induced Hemorrhage

We next analyzed whole fish for the clinically relevant signs of skin hemorrhage (Figure 8). The results indicated that at both time points, NCTL and HZFCTL (Figure 8a,b,g,h) did not show any skin hemorrhage whereas all DCTL fish receiving spike protein exhibited a significant increase (*p* < 0.001) in hemorrhage at the pelvic, anal, dorsal, and caudal fin regions (Figure 8c,i,m,n). Fish receiving either DEX (Figure 8d,j) or both doses of CN (Figure 8e,f,k,l) showed recovery from SARS-CoV-2 spike protein induced hemorrhage (Figure 8m,n), similar to untreated fish (NCTL).

### 2.10. Coronil Inhibits IL-1β Induced Secretion of IL-6 and TNF-α in Human Alveolar Epithelial Cells and Attenuates Transcriptional Activation of NF-κB/AP-1 Pathway

Cytokine storm in lungs is an important pathological feature of SARS-CoV-2 infection [35]. Therefore, we asked if Coronil could moderate the secretion of pro-inflammatory cytokines from human respiratory epithelial cells. We employed A549 cells that were also used for generating the humanized fish model. We first tested the cyto-safety of Coronil by measuring the cell viability of A549 cells. The results indicated no cytotoxicity of CN up to the test concentration of 300 μg/mL (Figure 9a). We tested for the IL-1β induced cytokine secretion and possible anti-inflammatory properties of CN (Figure 9b,c). Induction of A549 cells with 500 pg/mL IL-1β significantly enhanced the secretion of IL-6 and TNF-α (Figure 9b,c). Importantly, Coronil treatment resulted in a dose dependent statistically significant decrease (*p* < 0.001) in the secretion of both IL-6 (Figure 9b) and TNF-α (Figure 9c) in the IL-1β induced A549 cells.

Involvement of several transcription factors in lung inflammation and cytokine secretion has been well described including the master regulator, nuclear factor-kappaB (NF-κB), and activator protein-1 (AP-1) [36]. We tested for the inhibitory properties of CN on TNF-α induced NF-κB/AP-1 promoter transcriptional activation using the HEK-Blue^TM^ TNF-α reporter system. While bioactive TNF-α induced a 3.5-fold increase in NF-κB/AP-1 promoter transcriptional activity, pre-treatment of these reporter cells with Coronil significantly inhibited the TNF-α induced NF-κB/AP-1 transcriptional activation (Figure 9d) in a dose dependent manner. Collectively, these cell biological results confirm the anti-inflammatory properties of Coronil observed in the humanized zebrafish model and indicate a potential mode of action.

## 3. Discussion

The lack of specific treatment options to combat SARS-CoV-2 infection present an unprecedented challenge to identify novel drugs for prevention and treatment [37,38]. In the present original research, we were successful in demonstrating the disease-modulating properties with apparent therapeutic benefits of the herbal medicine, Coronil, on SARS-CoV-2 induced pathology. Furthermore, we also demonstrated that a translational dose of Coronil is effective in modifying various cellular and molecular endpoints of SARS-CoV-2 pathology using a physiologically relevant humanized zebrafish model. We also demonstrated the anti-inflammatory properties of Coronil in inhibiting IL-1β induced, IL-6, and TNF-α cytokine secretion in a NF-κB/AP-1 dependent manner in vitro, suggesting its immunomodulatory mode of action.

The clinical spectrum of SARS-CoV-2 infection ranges between asymptomatic forms to acute bilateral pneumonias, fever, fatigue and dry cough, Acute Respiratory Distress Syndrome (ARDS), and lymphopenia [7,35]. ARDS is a major devastating event in SARS-CoV-2 infection due to inflammatory injury to alveolo-capillary membrane pulmonary edema and hypoxemia [39]. This has been known to result in pulmonary and extra pulmonary capillary congestion, pneumocyte hyperplasia, necrosis of the hyaline membrane and pneumocytes, interstitial edema, and inflammatory cell accumulation in the lungs [40]. In the present study, SARS-CoV-2 spike protein challenge resulted in the induction of behavioral fever, and treatment with Coronil modestly decreased this induced behavioral fever. Furthermore, Coronil mediated inhibition of swim bladder collapse suggests the beneficial role of Coronil in rescue from edema. The extra pulmonary manifestations of SARS-CoV-2 infection include impairment of renal function, severe acute tubular necrosis, and inflammatory cell infiltration [41,42]. In this study, Coronil mediated recovery of SARS-CoV-2 spike protein injected in zebrafish kidneys suggests the role of Coronil in protecting against renal cell necrosis and degeneration.

In SARS-CoV-2 infection, innate immune response in the lungs leads to macrophage and monocyte recruitment, cytokine release, and initiate adaptive T and B cell immune responses for resolution of infection. However, in some cases, a dysfunctional immune response can result in severe lung and even systemic pathology [13]. In the presented research, spike protein challenge in DCTL fish induced accumulation of macrophages and granulocytes. More importantly, CN treatment induced the decrease in inflammatory cell accumulation and a varied lymphocytic accumulation may suggest the possible involvement of adaptive immune response in the rescue of the pathology.

In SARS-CoV-2 infection, granulocyte and monocyte abnormality, lymphopenia, and aberrant cytokine secretion have been noted [43]. Infiltration of polymorphic-nuclear neutrophils (PMN) in the airways during the early stages of respiratory viral infections including coronaviruses is evident [44,45] and their presence casually correlates with the clinical symptoms or severe disease pathology [46]. Furthermore, PMN exert antiviral activities and also activate innate and adaptive immunity, and thus contribute to effective antiviral responses [45,47]. Nevertheless, the complete role of PMNs in coronaviral infections is not fully understood [44]. Our results are in line with the previous reports as indicated by the presence of granulocytes only at the fourth day time point, which is the early stage of disease. The observed decrease in the SARS-CoV-2 spike protein induced granulocyte and macrophage infiltration upon Coronil treatment suggests the anti-inflammatory role of the ingredients present in Coronil. Lymphopenia is widely observed in SARS-CoV-2 infection [48] and absolute lymphocyte count has also been proposed as a biomarker for the severity of coronavirus infection with low levels of lymphocytes at the early stage of infection (immune dysfunction), indicating severe progress of disease [48,49]. In the present study, lymphocytes appeared only after the fourth day time point, indicating the activation of an adaptive immune system upon Coronil treatment and restoration of a number of lymphocytes might have played a role in scavenging the spike protein. Upon infection, SARS-CoV-2 spike protein becomes highly glycosylated, leading to masking of immunogenic epitopes from the host humoral immune system and evading the host immune system by occluding them with host-derived proteins [50]. We could not find any lymphocytes and macrophages in DCTL after the fourth day time point, which could be due to the masking of glycoprotein by the zebrafish host system.

A subgroup of patients with severe SARS-CoV-2 infection have “cytokine storm syndrome”, characterized by a hyper-cytokinemia associated with multi-organ failure [35]. SARS-CoV-2 infection of the respiratory tract causes the secretion of various pro-inflammatory cytokines including IL-1β, IL-6, and TNF-α [7,35,51]. The ACE2 receptor is highly expressed in alveolar epithelial type II cells (pulmonary pneumocytes) [52], suggesting the critical involvement of these cells. To simulate the response of alveolar epithelial cells, and the role of robust cytokine secretion in amplification and worsening of pathological features, alveolar type-2 A549 cells showed increased secretion of IL-6 and TNF-α cytokines upon stimulation with IL-1β. Importantly, treatment of these induced cells with Coronil attenuated the IL-6 and TNF-α cytokine secretion. Decreased T cell number in SARS-CoV-2 infection inversely correlates with TNF-α, IL-6, and IL-10 levels, suggesting that an increase in inflammatory cytokine levels can promote T cell depletion and exhaustion, leading to disease progression [53]. These results highlight the anti-inflammatory and immunomodulatory properties of Coronil.

Involvement of nuclear factor-kappaB (NF-κB) and activator protein-1 (AP-1) in orchestrating lung inflammation and cytokine secretion [48] has been well documented. Inhibitory activity of Coronil on the NF-κB/AP-1 pathway suggests the possible role of Coronil in inhibiting the robust cytokine secretion seen in the infection. NF-κB activation is mediated by ROS and PPARα, and PPARα is known to regulate immunity to the presence of DNA in the cytoplasm [54]. Several different naturally available ingredients have been shown to ameliorate the NF-κB activation seen in a host of metabolic disorders through the PPARα pathway [55,56]. SARS-CoV-2 is a cytopathic virus that infects airway epithelial cells and induces pyroptosis and vascular leakage [13]. SARS-CoV-2 recognition by Toll Like Receptor (TLR) causes the production of IL-1β, which mediates lung inflammation, fever, and fibrosis [51]. IL-6 is produced by various cells and is critically involved in regulating the acute phase inflammatory responses [57], and its production is increased by both IL-1β and TNF-α [58]. IL-1β also induces the expression of TNF-α [59], which is an important cytokine for systemic inflammation and for acute phase reaction. Both TNF-α and IL-6 signal the immune cells to induce inflammation [60]. TNF-α is a key cytokine in viral diseases and various chronic inflammatory and autoimmune diseases [61]. In the context of SARS-CoV-2 infection, TNF-α has been shown to facilitate SARS-CoV-2 interaction with ACE2, and subsequent entry of the virus into the host cells [62].

The anti-inflammatory and anti-viral properties of the herbal components of Coronil and their constituent phyto-compounds has been well documented. *T. cordifolia* has been used for its medicinal properties for thousands of years and contains alkaloids, glycosides, steroids, and polysaccharides [63]. *W. somnifera* has been known to enrich the physical and mental state and rejuvenate the body [64]. It has well-described anti-inflammatory, antimicrobial, analgesic, anti-stress, neuroprotective, cardioprotective, and immunomodulatory effects [65] and contains steroidal saponin, alkaloids, and steroidal lactones as bioactive ingredients [64]. The leaves of *O. sanctum* have been used for the treatment of rheumatism, bronchitis, and pyrexia and enhances cellular and humoral immunity [66]. UHPLC analysis of Coronil identified the presence of ursolic acid, betulinic acid, withanone, withaferine A, withanoside IV, withanoside V, cordifolioside A, magnoflorine, rosmarinic acid, and palmatine as bio-active molecules.

In line with our studies, molecular docking studies on *T. cordifolia* phyto-constituents to target the active site pocket of the 3CL^pro^, the main protease of SARS-CoV-2, identified that tinosponone, xanosporic acid, cardiofolioside, tembetarine, and berberine dock significantly to the 3CL^pro^ protease [67]. Further in silico studies have also confirmed that tinocordioside and berberine present in *T. cordifolia* may also regulate SARS-CoV-2 entry and its replication [68,69]. Magnoflorine, tinocordioside, syringin, and cordifolioside A are known for their immunomodulatory effect [70]. Magnoflorine and tinosporide isolated from *T. cordifolia* inhibits the HIV-1 protease active site with a very high affinity [71]. Palmatine inhibits respiratory syncytial virus in vitro and in vivo by inducing an antiviral state [72]. Furthermore, using the molecular docking approach, we and others have demonstrated the possible use of *W. somnifera* against SARS-CoV-2, wherein withanone, bound at the interface of the ACE2–RBD complex, and interrupted the electrostatic interactions between the RBD and ACE2. It has been postulated that withanone would either block or weaken SARS-CoV-2 entry into host cells and may inhibit the subsequent viral infectivity [73]. In addition, withanoside X and quercetin glucoside docked well within the targets NSP15 endoribonuclease and receptor binding domain of the spike protein, suggesting their potential as SARS-CoV-2 replication inhibitors [74]. It is also predicted that withanone and withaferin-A can interact with transmembrane protease serine 2 (TMPRSS2), thereby potentially could block the entry of SARS-CoV-2 into host cells [75]. A combination of withanone and caffeic acid phenethyl ester [76] or withanoside-V [77] has also been predicted to inhibit SARS-CoV-2 activity by interacting with its main protease (M^pro^). Anti-viral properties of ursolic acid [78] and betulinic acid [79] have been reported, with ursolic acid also inhibiting M^pro^ under in silico conditions [80]. Anti-inflammatory effect of rosmarinic acid has been credited with a reduction of acute lung injury in mice infected with the influenza virus [81]. Taken together, using in vivo and in vitro studies, our findings validate for the first time that the tri-herbal formulation Coronil has potent anti-inflammatory and immunomodulatory profile against SARS-CoV-2 spike protein induced pathologies.

## 4. Materials and Methods

### 4.1. Ethics Statement

Animal ethics guidelines from the Committee for the Purpose of Control and Supervision of Experiments (CPCSEA), Government of India, were followed to conduct the zebrafish (*Danio rerio*) experiments. Established zebrafish protocols were duly approved by the Institutional Animal Ethics Committee (IAEC study number-223/Go062020/IAEC).

### 4.2. Preparation of Coronil Sample for UHPLC

Coronil tablets were sourced from Divya Pharmacy, Haridwar, India (Batch number A-CNT112; expiry date June 2023). Tablets were crushed into fine powder and 0.5 gm of the powdered sample was diluted in 10 mL water:methanol (20:80) and sonicated for 30 min. The samples were centrifuged at 10,000 r.p.m. for 5 min and filtered using 0.45 µm nylon filter, and used for UHPLC analysis.

The quantification of signature compounds was performed using a Prominence-XR UHPLC system (Shimadzu Corporation, Kyoto, Japan) equipped with a quaternary pump (Nexera XR LC-20AD XR), DAD detector (SPD-M20 A), auto-sampler (Nexera XR SIL-20 AC XR), degassing unit (DGU-20A 5R), and column oven (CTO-10 AS VP). The separation was achieved using a Shimadzu Shim pack GIST-HP C18 (3 µm, 3 × 100 mm) (Shimadzu Corporation, Kyoto, Japan) column subjected to binary gradient elution using solvent-A (water containing 0.1% orthophosphoric acid, pH 2.5 adjusted with diethylamine) and solvent-B (acetonitrile). Gradient programming of the solvent system used was as follows: 5% solvent-B for 0–10 min, 5–15% solvent-B from 10–20 min, 15–25% solvent-B from 20–40 min, 25–65% solvent-B from 40–60 min, 65–90% solvent-B from 60–65 min, 90–5% solvent-B from 65–66 min, and 5% solvent-B from 66–70 min, with a flow rate of 0.7 mL/min. The following reference standards dissolved in methanol to prepare the appropriate concentrations used for the analysis. Palmatine hydrochloride (potency—97.0%, Sigma Aldrich, St. Louis, MO, USA), cordifolioside A (potency—98.6%, Chemfaces, Wuhan, Hubei, China), magnoflorine (potency—99.0%, Sigma Aldrich, St. Louis, MA, USA), withanone (potency—93.4%, Natural remedies, Mumbai, India), withaferine A (potency—98.0%, Natural remedies, Mumbai, India), withanoside IV (potency—94.3%, Natural remedies, Mumbai, India), withanoside V (potency—99.8%, Natural remedies, Mumbai, India), rosmarinic acid (potency—98.0%, Sigma Aldrich, St. Louis, MA, USA), ursolic acid (potency—97.3%, Tokyo chemical industries, Tokyo, Japan), and betulinic acid (potency—98.0%, Natural remedies, Mumbai, India). 10 µL of the standard and test solution were injected and the column temperature was maintained at 30 °C. The chromatograms were recorded at 210 nm (for ursolic acid and betulinic acid), 227 nm (for withanone, withaferine A, withanoside IV, withanoside V, cordifolioside A and magnoflorine), and 325 nm (for rosmarinic acid and palmatine) wavelengths.

### 4.3. Zebrafish Care and Maintenance

Six groups of 24 adult wild-type zebrafish in each group were maintained in a dedicated zebrafish research facility and followed the standard photoperiod of a 14 h light, 10 h dark cycle. The water temperature was maintained at 27 ± 1 °C. Zebrafish of similar bodyweight (0.5 g), weighed in a smaller beaker with a known amount of water, were selected for the experimental study and were housed in polycarbonate tanks at a stocking density of 2 L of water per fish. The fish under housing conditions were fed once per day with 2.5 mg feed per gram body weight of fish using commercial feed (TetraBit, Spectrum Brands Pet LLC, Blacksburg, VA, USA).

### 4.4. Preparation of Zebrafish Test Feed and Dosing

To prepare the infused oral feeds, the study compound, powdered Coronil, and Dexamethasone were diluted with PBS according to the translation dose for zebrafish [82,83]. The required volume of Coronil and DEX were mixed and ground with a known quantity of fish feed and extruded to pellets of uniform size, weighing 2.5 mg per pellet for oral dosing. A 24 h feeding cycle with the estimated number of pellets per fish were followed throughout the study period (three and six days). Test article information for each study group was blind coded for investigators and fish-handlers. For the control group, fish feed was mixed with an equal volume of PBS without drug and extruded. During the dosing session, the fish were isolated from the respective study groups and were fed individually in the feeding tank. The feeding tank is a rectangular tank divided into six independent units. Control fish were fed with unmodified fish feed and were fed under similar conditions to that of the study groups.

### 4.5. Xeno-Transplantation of A549 Cells into Humanized Zebrafish Using Intramuscular Injection

Human alveolar epithelial (A549) cells from highly confluent cultures from the third passage were diluted in PBS for xeno-transplantation (2 × 10^2^ cells per fish) and injected intramuscularly to seed A549 cells at the posterior lobe of zebrafish swim bladder. The fish from the clutches were isolated and anesthetized using gradual cooling of the water with ice cubes. The fish were gently transferred from the clutches to the first anesthesia tank maintained at 17 °C and were observed for gradual decrease in the operculum movement and upon decrease in the operculum movement, the fish were transferred to 12 °C until the fish were ready to handle as confirmed by the absence of response to caudal fin touch. The anesthetized fish were placed on the injection stage and the injection needle was inserted to the region at the junction of the trunk and the caudal region, in the midline between the dorsal and ventral side as this was the closest region to the posterior lobe of the swim bladder. Post injection, the fish were transferred to the respective clutches and were observed for a period of seven days. Cytology of the swim bladder was studied on day 7 to confirm the A549 cell adherence to the swim bladder of zebrafish.

### 4.6. Induction of Pathology Milieu of SARS-CoV-2 Infection in Zebrafish

Post the 7-day period after xeno-transplantation, the fish were injected with the recombinant SARS-CoV-2 spike protein (Bioss, Woburn, MA, USA). To prepare the injection mixture, spike protein stock solution was prepared by dissolving 100 µg of SARS-CoV-2 spike protein in 100 μL of PBS to make a final concentration of 1 μg/µL. The working solution was prepared by dissolving 1 µL of the stock to a final volume of 1 mL of PBS to arrive at a concentration of 1 ng/µL. 2.8 µL (2.8 ng) of the SARS-CoV-2 spike protein working solution was loaded to a Hamilton syringe and injected carefully to the site of xeno-transplant at the junction of the trunk and the caudal region, along the midline. Post xeno-transplantation and recombinant spike protein injection the zebrafish were housed at a temperature of 27 ± 1 °C to aid the physiological changes in immunity.

### 4.7. Survival Tests

Survival and mortality of zebrafish in all groups were counted every day and evaluated using Kaplan-Meier survival curves. For each fish, time of death, survival status, and group information were recorded and then Kaplan-Meier survival tests were run to generate the survival curve. *p*-values were calculated using the log-rank test.

### 4.8. Behavioral Fever Assessment

To study the behavioral fever, an experimental glass tank was set up with a variation of three temperatures (23, 29, and 37 °C) gradient interconnected chambers (Figure 10). The tank temperature was maintained with continuous heating and cooling from either side with a hot bath at 40 °C and cooling at 18 °C to maintain the temperature during the study period [84]. The fish from the respective study clutches were introduced individually into the temperature gradient chamber at 29 °C, thereby giving an opportunity to choose a temperature that supported the physiological state of the fish. The time spent by the fish in the temperature gradient chamber was recorded for a consecutive 3 min time period.

### 4.9. Examination for Presence of Skin Hemorrhage

At the time of termination, fish were euthanized, and images were captured on a glass slide before dissection. Whole animal imaging was performed using a digital single lens reflex (DSLR) camera (Nikon D3100, Nikon Corporation, Chiyoda, Japan) for demarking the features that represent phenotypic variations. The area of hemorrhage was measured using NIH ImageJ image software, version 2 (Laboratory for Optical and Computational Instrumentation, University of Wisconsin, Madison, WI, USA). The polygonal tool was used to mark the outline of the hemorrhagic area and was measured (in mm^2^) using the image analysis tool.

### 4.10. Harvesting Swim Bladder and Kidney for Anatomical Observations

On the last day of the experiment, the fish were euthanized using cold water and dissected as per ethical guidance. Individual fish were pinned to a dissection board with the ventral side up and dissected through a ventral incision in the skin from the lower jaw to the vent. The heart, intestinal cavity, and gonads were removed to expose the swim bladder and kidney. Both were carefully excised without any damage and observed under the Labomed CM4 stereomicroscope (Los Angeles, CA, USA) at 1X magnification using a Labomed camera (Los Angeles, CA, USA).

### 4.11. Cytology of Swim Bladder and Kidney

Cytosmears were prepared from the whole swim bladder and kidney necropsy and were stained with hematoxylin and eosin (H&E) for 2 min each, followed by three PBS washes and then allowed to dry at room temperature. Slides were viewed under a Labomed light microscope (Los Angeles, CA, USA) at 200X magnification. For kidney smear analysis, the average degenerative and necrotic nuclei per three field were quantified. The leukocytes were identified by their nuclear morphology and were quantified as an average number of cells present in three different fields under microscopy.

### 4.12. In Vitro Cell-Biology Assays

#### 4.12.1. Cell Culture

Human lung alveolar adenocarcinoma cell line, A549 (purchased from ATCC licensed cell repository, National Center for Cell Sciences (NCCS), Pune, India) were cultured in DMEM (Dulbecco’s modified Eagle’s medium; Invitrogen, Carlsbad, CA, USA) supplemented with 10% FCS and antibiotics. HEK-Blue™ TNF-α cells (InvivoGen, San Diego, CA, USA) were maintained in complete DMEM containing l-glutamine (2 mM), 100 U/mL penicillin, 100 μg/mL streptomycin, 100 μg/mL normocin, and 10% heat inactivated FBS. Cells were placed in a humidified incubator containing 5% CO_2_ at a temperature set at 37 °C. For in vitro cell biology experiments, cells were used at 70% confluency.

#### 4.12.2. A549 Cell Viability Assay

A549 cell viability was measured using Alamar blue reagent (Hi Media Laboratories, Mumbai, India). Briefly, A549 cells (3 × 10^4^ cells/well) were seeded in 96-well plates and treated for 24 h with various concentrations of Coronil (1–1000 µg/mL). Two hours before termination, 10 µL of Alamar blue (0.15 mg/mL) was added to each well. Cytotoxicity was measured by reading fluorescence at Ex. 540/Em. 590 using the Envision microplate reader (Perkin Elmer, Waltham, MA, USA).

#### 4.12.3. In Vitro Anti-Inflammatory Activity of Coronil and Quantification of IL-6 and TNF-α Cytokines

The A549 cells were pre-treated with various dilutions of Coronil (1–100 µg/mL) for 24 h followed by co-treatment with Coronil + 500 pg/mL human IL-1β for another 24 h. Cell culture supernatant was collected and used for measuring the secretion of cytokines. Enzyme-linked immunosorbent assay (ELISA) of cytokine levels in A549 cell culture supernatants were analyzed using human IL-6 and TNF-α (BD Bioscience, San Jose, CA, USA) according to the manufacturer’s instructions and the ELISA plates were read at 450 nm using an Envision microplate reader (Perkin Elmer, Waltham, MA, USA).

#### 4.12.4. Secreted Embryonic Alkaline Phosphatase (SEAP) Based NF-κB/AP-1 Reporter Assay

HEK-Blue™ TNF-α cells were resuspended in 3 × 10^5^ cells/ mL and plated in 96-well plates at 100 μL/well in complete DMEM medium without Normocin. Serial dilutions of Coronil (1 μg–100 μg) were added to cells and incubated at 37 °C, 5% CO_2_ for 12 h followed by the addition of 500 pg/mL of human TNF-α. After 24 h of incubation, 20 μL of supernatants were incubated for 1 h with 180 μL of QUANTI-Blue™ reagent to detect secretion of alkaline phosphatase according to the manufacturer’s protocol (InvivoGen, San Diego, CA, USA); plates were read at 630 nm using an Envision microplate reader (Perkin Elmer, Waltham, MA, USA).

### 4.13. Statistical Analysis

All in vivo experiments were conducted with multiple replicates (*n* = 24 for each test group), in vitro experiments were conducted in triplicate, and data are expressed as means ± SEM. Comparisons between experimental groups were made by one-way ANOVA, followed by Dunnett’s multiple comparison post-hoc test. Differences in mean values were considered significant at *p* < 0.05. Significance of data were analyzed using GraphPad Prism version 7.04 (GraphPad Software LLC, San Diego, CA, USA).

## 5. Conclusions

In the present study, we successfully established a model for SARS-CoV-2 spike protein induced disease phenotype in zebrafish and validated the disease rescue properties of an Indian tri-herbal medicine, Coronil, at human equivalent doses. The anti-inflammatory properties were determined using various cellular and molecular endpoints. Use of Coronil as a complementary medicine during SARS-CoV-2 infection may alleviate disease pathological features by acting as an immuno-modulator. Results obtained from the present preclinical study warrants detailed clinical trials of Coronil in human subjects of SARS-CoV-2 infection.

## Figures and Tables

**Figure 1 molecules-25-05091-f001:**
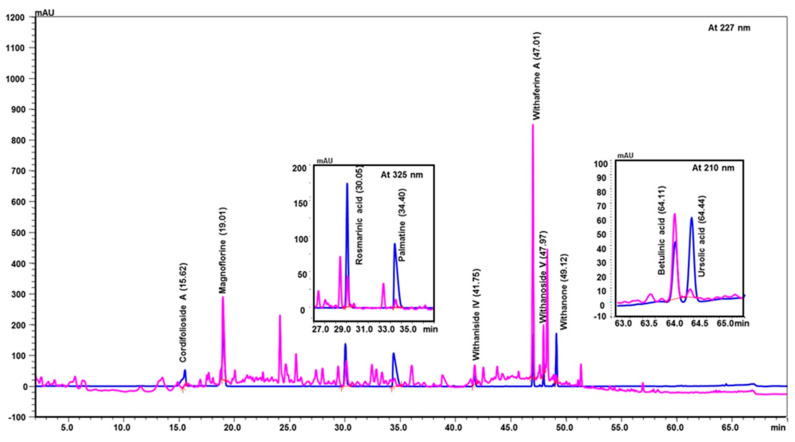
UHPLC analysis identified the presence of bio-active ingredients in Coronil. The Coronil sample (purple lines) was crushed into a fine powder and analyzed by UHPLC using reference standards mix (blue lines). The chromatographs were recorded at 210 nm (for betulinic acid and ursolic acid), 227 nm (for withanone, withaferine A, withanoside IV, withanoside V, cordifolioside A and magnoflorine), and 325 nm (for rosmarinic acid and palmatine) wavelengths. The chemical structures of these identified compounds, retention time (in minutes), concentration (µg/mg) and the plant species it was derived from are given in Table 1.

**Figure 2 molecules-25-05091-f002:**
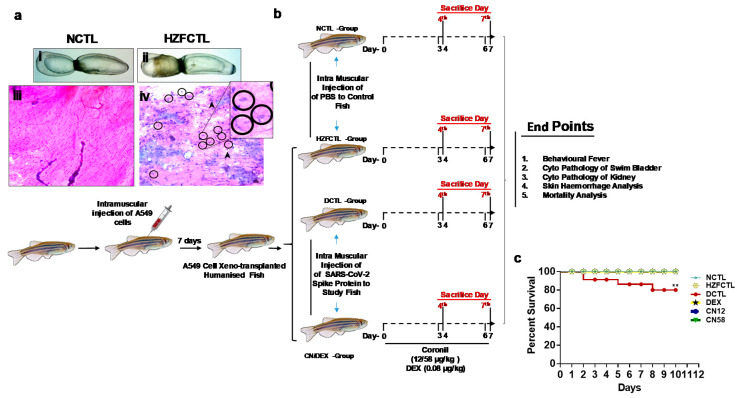
Study design of SARS-CoV-2 spike protein induced pathology using humanized xeno-transplanted zebrafish. (**a**) Representative images of swim bladders from (**i**) NCTL and (**ii**) HZFCTL post 7-day transplant showing normal morphological traits. Cyto-smears of (**iii**) NCTL and (**iv**) HZFCTL showed normal cytological features along with transplanted cells (zoom-in represents xeno-transplanted cells). Black arrows indicate cells of zebrafish origin and black circle represent cells of human origin. AL—anterior lobe, PL—posterior lobe. Scale bars, 1000 μm at 1X magnification. (**b**) Schematic representation of zebrafish model of SARS-CoV-2 spike protein induced pathology. Humanized zebrafish (*n* = 24 in each group) were injected with SARS-CoV-2 spike protein (Disease Control, DCTL) or PBS (Normal Control, NCTL and Humanized zebrafish control, HZFCTL) and treated with two different doses of Coronil, 12 μg/kg (CN12) and 58 μg/kg(CN58) and DEX (0.08 μg/kg) for two time points followed by necropsy and endpoint measurement. (**c**) Kaplan-Meier survival curves following SARS-CoV-2 spike protein challenge and pharmacological treatment with DEX and Coronil till 10th day. *n* = 24. Log rank test ** *p* < 0.001.

**Figure 3 molecules-25-05091-f003:**
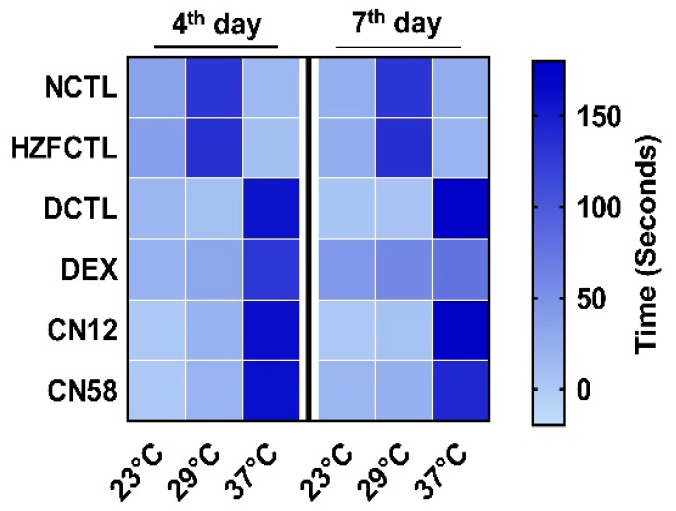
Coronil reduces SARS-CoV-2 spike protein induced behavioral fever. Heat map of time spent (in seconds) at the set temperatures (low temperature 23 °C, normal temperature 29 °C, and behavioral fever 37 °C) by the zebrafish from various groups on the fourth and seventh day time points. Color intensity indicates the time spent at the particular temperature chamber. SARS-CoV-2 spike protein induced behavioral fever was decreased in a dose dependent manner upon CN treatment at the seventh day time point.

**Figure 4 molecules-25-05091-f004:**
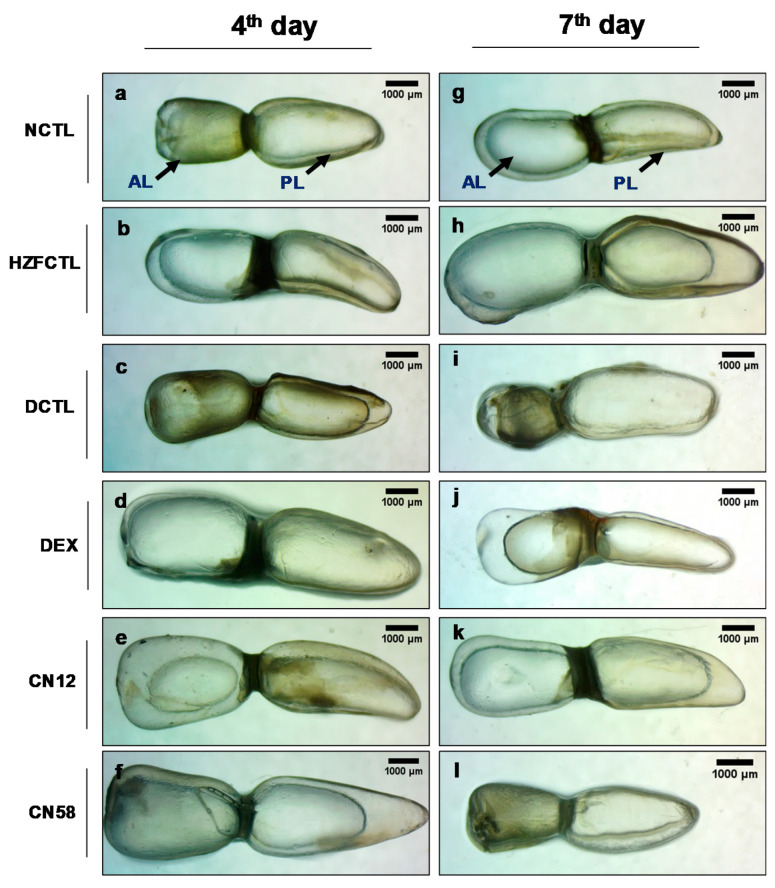
Coronil reduces SARS-CoV-2 induced swim bladder structural abnormality and edema. (Normal control at 4th day (**a**), HZFCTL at 4th day (**b**) and NCTL at 7th day (**g**), HZFCTL at 7th day (**h**)) NCTL and HZFCTL at the 4th and 7th day time points, respectively, showing normal swim bladder. (**c**,**i**) SARS-CoV-2 spike protein induced structural anomaly and edema. (**d**) DEX treatment at the 4th day time point showed mildly inflated swim bladder. (**j**) DEX treatment at the 7th day time point showed narrowing of the posterior lobe due to edema. (**e**,**f**) At the 4th day time point, the CN12 and CN58 groups were comparable to the NCTL and HZFCTL groups. (**k**,**l**) At the 7th day time point, the CN12 and CN58 groups were comparable to NCTL. *n* = 24 for each test group, AL—anterior lobe, PL—posterior lobe. Circle inside the swim bladder indicates the watermark at the contact between the glass slide and swim bladder during the image capturing process. Scale bar, 1000 μm at 1X magnification.

**Figure 5 molecules-25-05091-f005:**
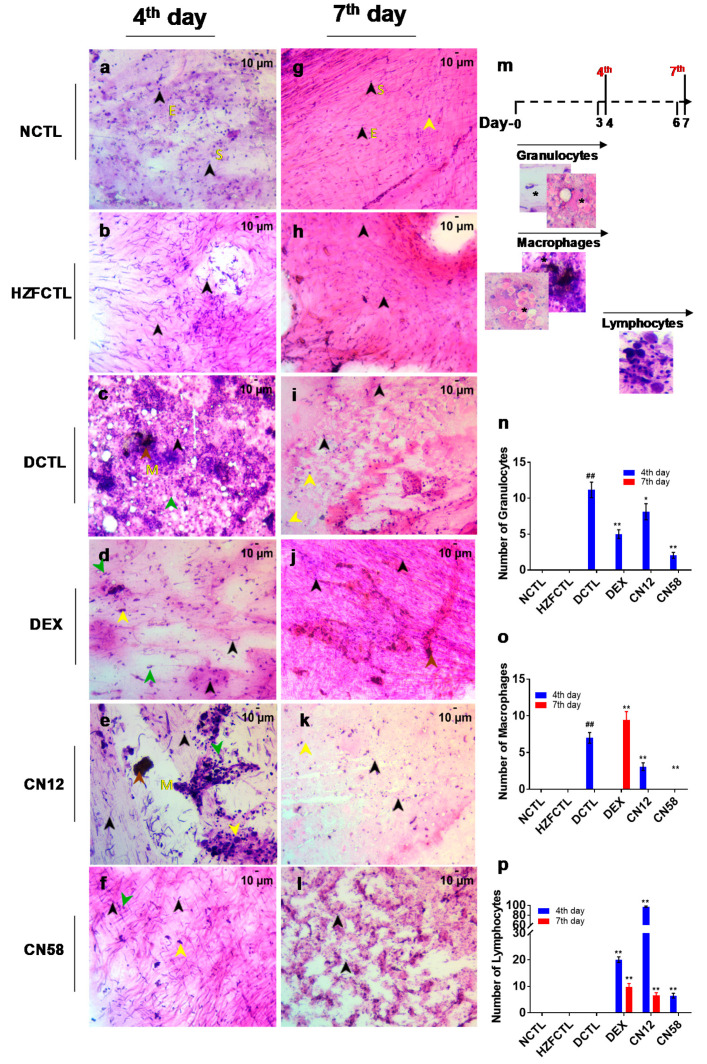
Coronil rescues from SARS-CoV-2 spike protein induced inflammation in swim bladder cytosmears. Cytosmears of the swim bladder were analyzed for inflammatory cell accumulation based on cell and nuclear morphology. Representative images of Hematoxylin and Eosin (H&E) stained swim bladder cytosmears of (**a**) NCTL at 4th day and (**g**) NCTL at 7th day and HZFCTL at 4th day (**b**) and at 7th day (**h**) showed cells with normal morphological characteristics. (**c**,**i**) DCTL showed infiltration of granulocytes and macrophages and cellular materials and nuclear debris and edema. (**d**,**j**) DEX treatment rescued from inflammation along with the presence of cellular debris. (**e**) Coronil (12 µg/kg) at 4th day, (**f**) Coronil (58 µg/kg) at 4th day; (**k**) Coronil (12 µg/kg at 7th day, and (**l**) Coronil (58 µg/kg at 7th day. Dose dependent rescue of cytological disease markers on both the 4th day (**e**—CN12, **f**—CN58) and 7th day (**k**—CN12, **l**—CN58) time points upon Coronil treatment as indicated by the presence of epithelia and myocytes along with lymphocyte and macrophage aggregates and reduced inflammatory cellular debris. (**n**) Quantitative enumeration of granulocytes using swim bladder cyto-smear. (**o**) SARS-CoV-2 induced macrophage infiltration was significantly decreased upon CN treatment at the 4th day time point. (**p**) Lymphocyte infiltration was significantly increased upon DEX or CN treatment at both time points. Scale bars, 10 μm at 200X magnification. *n* = 24 for each test group. ## represents statistical significance compared to HZFCTL and ** represents significant compared to DCTL (*p* < 0.001). E—epithelial nuclei; S—smooth muscle cell nuclei; M—melano-macrophage. Black arrow—normal cell, yellow arrow—lymphocyte, brown arrow—macrophage and green arrow—granulocyte.

**Figure 6 molecules-25-05091-f006:**
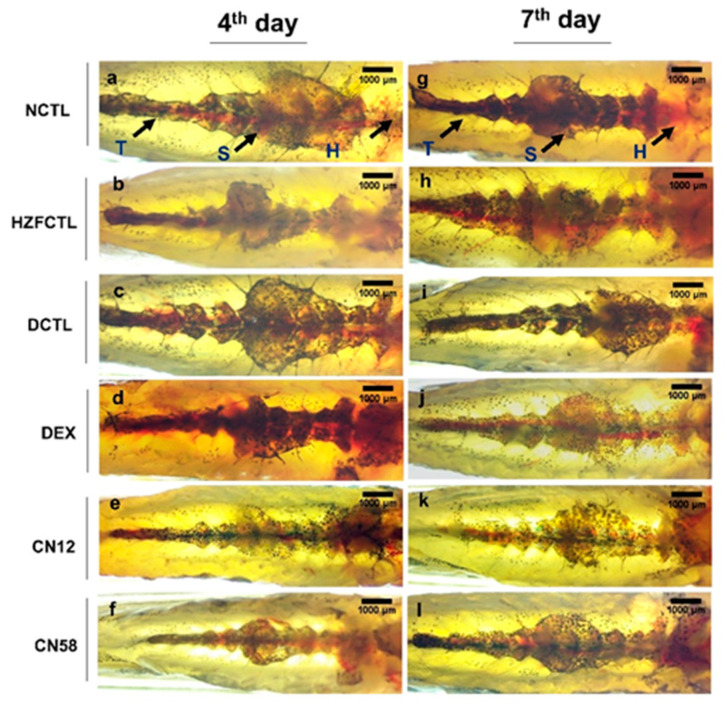
Gross anatomy of kidney to test the effect of SARS-CoV-2 spike protein and Coronil on renal architecture. (**a**) NCTL at 4th day, (**b**) HZFCTL at 4th day, (**g**) NCTL at 7th day, (**h**) HZFCTL at 7th day. (**c**,**i**) DCTL with lost tubular segmentation and vascular degeneration. (**d**,**j**) DEX. (**e**,**f**) CN12 and CN58 of 4th day group exhibiting normal renal morphology. (**k**) CN12 showed normal renal architecture at the 7th day time point. (**l**) CN58 exhibiting mildly degenerative renal pathology. H—Head; S—Saddle; T—Tail of kidney. *n* = 24 for each test group, scale bars, 1000 μm at 1X magnification.

**Figure 7 molecules-25-05091-f007:**
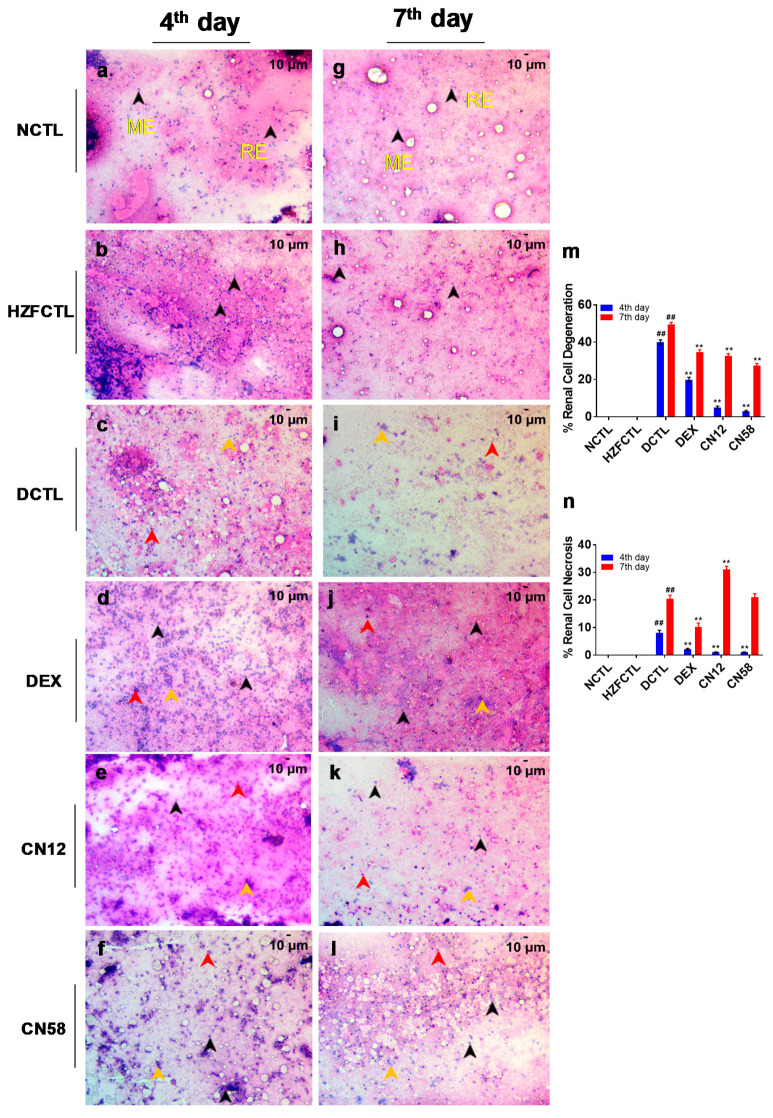
Coronil rescues from SARS-CoV-2 spike protein induced renal cell damage. Cytosmears of whole kidney necropsies were made and stained with H&E. The average degenerative and necrotic nuclei per 3 field were quantified to study renal pathological features at both the 4th and 7th day time points. (**a**) NCTL at 4th day, (**b**) HZFCTL at 4th day, (**g**) NCTL at 7th day, (**h**) HZFCTL at 7th day. NCTL and HZFCTL with apparent cytological features without any necrotic and degenerating cells. (**c**,**i**) DCTL with necrotic and degenerating renal cells. (**d**,**j**) DEX group. (**e**) Coronil (12 µg/kg) at 4th day, (**f**) Coronil (58 µg/kg) at 4th day; (**k**) Coronil (12 µg/kg at 7th day, and (**l**) Coronil (58 µg/kg at 7th day. CN12, CN58 at 4th day and 7th day time points. (**m**) Renal cell degeneration was significantly decreased in a dose dependent manner at the 7th day time point. (**n**) Spike protein induced renal cell necrosis % was significantly decreased at the 4th day time point. Black arrow—normal cell, Red arrow—necrotic cell and Yellow arrow—degenerating cell. Scale bars, 10 μm at 200X magnification. *n* = 24 for each test group. ## represents statistical significance compared to HZFCTL and ** represents significance compared to DCTL (*p* < 0.001).

**Figure 8 molecules-25-05091-f008:**
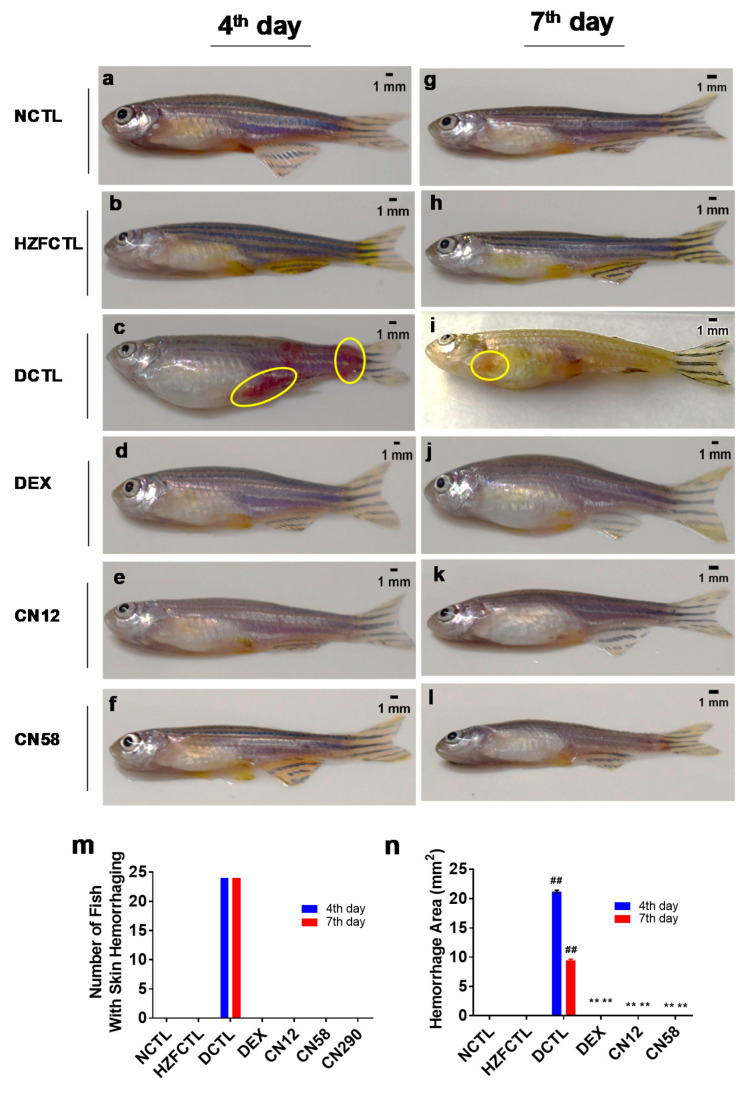
Pharmacological treatment with Coronil inhibits SARS-CoV-2 spike protein induced hemorrhage. Zebrafish were euthanized at end of study and whole animal imaging was performed from all groups and analyzed for spike protein induced skin hemorrhage on the 4th day and 7th day time points. (**a**) NCTL at 4th day, (**b**) HZFCTL at 4th day, (**g**) NCTL at 7th day, (**h**) HZFCTL at 7th day. (**c**,**i**) DCTL showing skin hemorrhage circled in yellow. (**d**,**j**) DEX group. (**e**) Coronil (12 µg/kg) at 4th day, (**f**) Coronil (58 µg/kg) at 4th day; (**k**) Coronil (12 µg/kg at 7th day, and (**l**) Coronil (58 µg/kg at 7th day. (**m**) Qualitative analysis of the number of fish with hemorrhage across each treatment group and respective time points. (**n**) Quantitative analysis of area of hemorrhage. *n* = 24 for each group, ## represents statistical significance compared to HZFCTL and ** represents significance compared to DCTL (*p* < 0.001).

**Figure 9 molecules-25-05091-f009:**
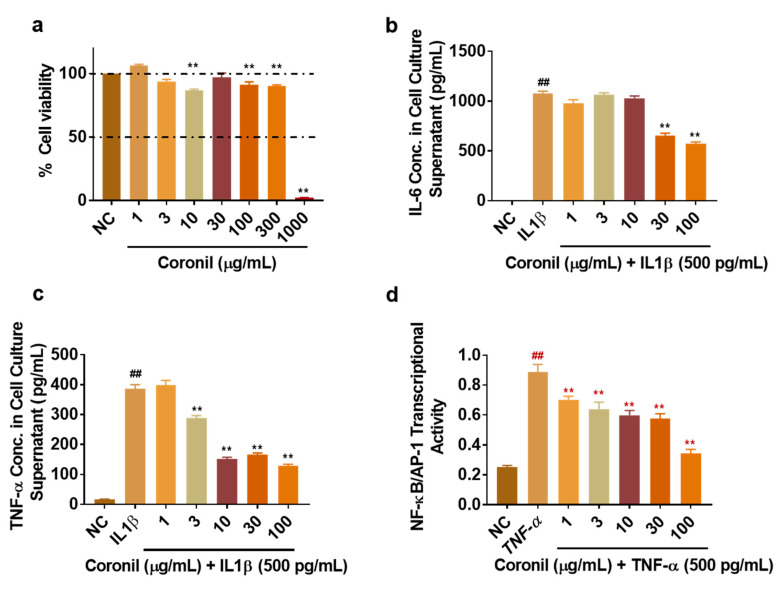
Coronil decreases the pro-inflammatory responses in vitro. A549 cells were treated with various concentrations of CN for 24 h and measured for toxicological profile. (**a**) Cytotoxicity of CN on A549 cells expressed as % cell viability. Cells were pre-treated with CN followed by co-treatment with CN + IL-1β for 24 h and assayed for the secretion of cytokines using Enzyme Linked Immunosorbent Assay (ELISA). Secretion of (**b**) IL-6 and (**c**) TNF-α. (**d**) HEK Blue TNF-α reporter Secreted Alkaline phosphatase (SEAP) assay for NF-κB/AP-1 transcriptional activity. HEK blue cells stably expressing the NF-κB/AP-1-SEAP reporter gene were pre-treated with various doses of CN followed by treatment with TNF-α and measuring SEAP activity after 24 h. Data are presented as the means ± SEM (*n* = 2 in triplicates). **, ^##^, *p* < 0.001 by one-way ANOVA; ^##^ represents significance compared to NC and ** represents significance compared to IL-1β.

**Figure 10 molecules-25-05091-f010:**
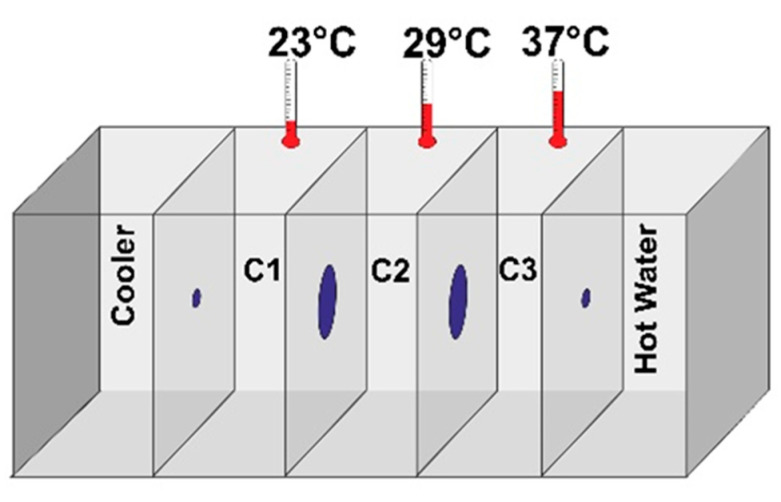
Schematic representation of the experimental tank setup with a variation of three temperature (23, 29 and 37 °C) gradient, interconnected chambers for studying behavioral fever in a study of zebrafish.

**Table 1 molecules-25-05091-t001:** Ultra-high-performance liquid chromatography (UHPLC) analysis of Coronil phyto-metabolites, as per the chromatograms shown in Figure 1.

Peak No.	Name of Phyto-Metabolites	Structure(as per PubChem Database)	Retention Time (min.)	Quantity (µg/mg)	Plant Species of Origin
1	Cordifolioside A	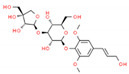	15.64	0.080	*Tinospora cordifolia*
2	Magnoflorine	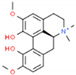	19.03	1.041	*Tinospora cordifolia*
3	Rosmarinic acid	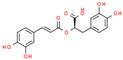	30.05	0.233	*Ocimum sanctum*
4	Palmatine	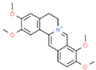	34.40	0.071	*Tinospora cordifolia*
5	Withanoside IV	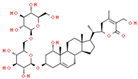	41.66	1.870	*Withania somnifera*
6	Withaferine A	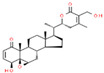	46.98	4.891	*Withania somnifera*
7	Withanoside V	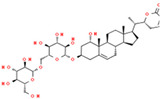	47.94	2.072	*Withania somnifera*
8	Withanone	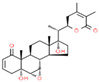	49.08	0.113	*Withania somnifera*
9	Betulinic acid	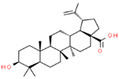	64.11	1.270	*Ocimum sanctum*
10	Ursolic acid	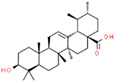	64.44	0.082	*Ocimum sanctum*

**Table 2 molecules-25-05091-t002:** Time (in seconds) spent by experimental fish of each test group in various temperature chambers. Data are expressed as mean ± S.D. values (*n* = 24 for each group).

Study Groups	Time (in Seconds) Spent in the Temperature Gradient Chamber					
	4th Day			7th Day		
	23 °C	29 °C	37 °C	23 °C	29 °C	37 °C
**NCTL**	35.58 ± 3.0	129.63 ± 1.4	14.92 ± 2.7	24.13 ± 0.7	130.00 ± 0.8	25.88 ± 0.7
**HZFCTL**	37.79 ± 1.5	134.00 ± 1.5	8.21 ± 2.0	25.88 ± 1.3	135.75 ± 1.5	18.38 ± 1.3
**DCTL**	15.96 ± 1.6	7.04 ± 1.3	157.08 ± 1.6	2.38 ± 1.1	4.75 ± 1.3	172.88 ± 1.3
**DEX**	21.13 ± 2.3	30.46 ± 2.3	128.29 ± 1.9	45.54 ± 1.6	57.54 ± 2.0	76.92 ± 2.1
**CN12**	0	19.08 ± 2.1	160.92 ± 2.1	0	5.04 ± 1.7	174.96 ± 1.7
**CN58**	0	19.92 ± 2.1	160.08 ± 2.1	17.54 ± 1.7	21.50 ± 1.8	140.96 ± 2.0

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
