# Peer review of "Application of Humanized Zebrafish Model in the Suppression of SARS-CoV-2 Spike Protein Induced Pathology by Tri-Herbal Medicine Coronil via Cytokine Modulation"

_molecules, 2020, doi:10.3390/molecules25215091_

Round 1

Reviewer 1 Report

Dear Author

I have only one question. A549 cells are a type of human lung cancer.

I would like to put the reference for the rational of A549 cells in human diseases model.

It is only my request.

Thank you.    

Author Response

I have only one question. A549 cells are a type of human lung cancer.

I would like to put the reference for the rational of A549 cells in human diseases model.

It is only my request.

Thank you.   

Response: Thank you for the kind suggestion.  We have now added the relevant references that discuss the use of A549 cells as a disease model in COPD and P. aeruginosa infection. Also, the study by Shen et al. (2020) that used A549 xeno-transplanted zebrafish to model lung cancer has also been added in the introduction.

This addition makes the introduction much clearer and robust. We appreciate your kind thoughts.

Reviewer 2 Report

The manuscript molecules-954085 is devoted the actual field of the pharmaceutical science, namely new possible anti-COVID drug and can be interested to the specialists working in this field. Work is performed at sufficient scientific level and has good quality. However, it needs major revision before publication.

To improve the quality and perception of the manuscript I would suggest paying attention to following comments:

To improve the quality and perception of the manuscript I would suggest paying attention to following comments:

  1. The style of some references should be changed. There are 4-5 sources after one sentence. This is unacceptable for publications in high-rated journals. Instead such references, it would be better to make a cross-reference discussion.
  2. The manuscript should be carefully reviewed for repetitions. In particular, the phrase "Coronil consists of enriched extracts of Tinospora cordifolia (Willd.) Miers, Ocimum sanctum L and Withania somnifera (L.) Dunal plants. The phytochemical analysis [44][45], the immunomodulatory [35][36][37][38][39] and anti-viral properties [40][41][42][43] of these three plants have been studied in-silico [35], in-vitro and in-vivo approaches." occurs twice (lines 103 and 502).
  3. Moderate English changes required. There are grammar/typing and orthographical errors in the manuscript.

My decision is minor revision.

Author Response

The manuscript molecules-954085 is devoted to the actual field of the pharmaceutical science, namely new possible anti-COVID drug and can be interested to the specialists working in this field. Work is performed at a sufficient scientific level and has good quality. However, it needs major revision before publication.

To improve the quality and perception of the manuscript I would suggest paying attention to the following comments:

1. The style of some references should be changed. There are 4-5 sources after one sentence. This is unacceptable for publications in high-rated journals. Instead such references, it would be better to make a cross-reference discussion.

Response: Thank you very much for your appreciative comments on our work. This definitely motivates us to go the extra mile in our ongoing experiments.  

As per the advice from the esteemed reviewer, changes have been made to include minimal references in one sentence. The relevant references have also been moved to various discussion points, as suggested.

2. The manuscript should be carefully reviewed for repetitions. In particular, the phrase "Coronil consists of enriched extracts of Tinospora cordifolia (Willd.) Miers, Ocimum sanctum L and Withania somnifera (L.) Dunal plants. The phytochemical analysis [44][45], the immunomodulatory [35][36][37][38][39] and anti-viral properties [40][41][42][43] of these three plants have been studied in-silico [35], in-vitro and in-vivo approaches." occurs twice (lines 103 and 502).

Response: Our apologies for this hindsight. The manuscript has now been reviewed extensively, as suggested and care has been taken to avoid verbatim repetition of information. Those specific phrases have also been edited and corrected.

3. Moderate English changes required. There are grammar/typing and orthographical errors in the manuscript.

Response: Thank you for pointing out a need for language correction. Changes have been made to correct language and a few typo errors, throughout the manuscript.

With these edits and suggestions, the manuscript does read much better. and has a smoother flow of information. Thank you very much.

Reviewer 3 Report

The manuscript is interesting and novel. The methodology is updated and consistent with the objective of the study. The results support the discussion, and the discussion includes the most relevant aspects. However, I have the following comments.

I. Major Comments:
1. The introduction is too long. In the introduction, the authors address aspects that should be included in the discussion. 46 references in the introduction is too much. The introduction should have a maximum of 20 to 25 references. It is important to modify the introduction.

2. The discussion is good, but I suggest:
2.1. The expression and activity of the transcription factor NF-kB is regulated by oxidative stress. Furthermore, the activity of NF-kB is dependent on PPAR-alpha. This interaction is very important, because it allows us to understand the interaction between the transcription factors. Discuss this point.

Suggested references:
Hydroxytyrosol supplementation ameliorates the metabolic disturbances in white adipose tissue from mice fed a high-fat diet through recovery of transcription factors Nrf2, SREBP-1c, PPAR-γ and NF-κB. Biomed Pharmacother. 2019; 109: 2472-2481.

Supplementation with Docosahexaenoic Acid and Extra Virgin Olive Oil Prevents Liver Steatosis Induced by a High-Fat Diet in Mice through PPAR-α and Nrf2 Upregulation with Concomitant SREBP-1c and NF-kB Downregulation. Mol Nutr Food Res. 2017; 61.

3. Considering the results, I suggest including a figure that summarizes the main results.

4. Many sentences are very long, I suggest editing the wording.

II. Minor comments:
1. Improve the writing of the study objective.

Author Response

Major Comments:

  1. The introduction is too long. In the introduction, the authors address aspects that should be included in the discussion. 46 references in the introduction is too much. The introduction should have a maximum of 20 to 25 references. It is important to modify the introduction.

Response: The esteemed reviewer’s comments about the length and content of the introduction are very valid. However, we intended to cover most of the current relevant work on SARS-Cov-2, therefore the reference numbers were on the higher side. As per the advice, we have now trimmed the introduction considerably; and the manuscript has been edited accordingly.

2.   The discussion is good, but I suggest:

2.1. The expression and activity of the transcription factor NF-kB is regulated by oxidative stress. Furthermore, the activity of NF-kB is dependent on PPAR-alpha. This interaction is very important, because it allows us to understand the interaction between the transcription factors. Discuss this point.

Response: The reviewer’s expert comments on the role of PPARα in NFkB regulation have been acknowledged, and included in the discussion along with the following advised references.

  • Hydroxytyrosol supplementation ameliorates the metabolic disturbances in white adipose tissue from mice fed a high-fat diet through recovery of transcription factors Nrf2, SREBP-1c, PPAR-γ and NF-κB. Biomed Pharmacother. 2019; 109: 2472-2481.
  • Supplementation with Docosahexaenoic Acid and Extra Virgin Olive Oil Prevents Liver Steatosis Induced by a High-Fat Diet in Mice through PPAR-α and Nrf2 Upregulation with Concomitant SREBP-1c and NF-kB Downregulation. Mol Nutr Food Res. 2017; 61.
  1. Considering the results, I suggest including a figure that summarizes the main results.

Response: This is a fantastic suggestion from the esteemed reviewer. We have generated a video abstract of this work that would visually summarise the results. We would upload this video abstract on the Molecules website, as per the options provided.

  1. Many sentences are very long, I suggest editing the wording.

Response: Changes have been made to make the manuscript crisp and of improved clarity. The long statements have been broken down into smaller ones.

  • Minor comments:
    Improve the writing of the study objective.

Response: Thank you for the suggestion. Changes have been made to make the study objective more relevant and to reflect the work presented.

Reviewer 4 Report

The authors present a study dedicated to the evaluation of a tri-herbal medicine (Coronil) against induced SARS-CoV-2 infection in humanized zebrafish model by evaluating different cellular, molecular, and physiological parameters. The overall conclusion was that Coronil was able to attenuate the inflammatory response and rescue the pathological features of SARS-CoV-2 suggesting its potential therapeutic use. Overall, this is an interesting, new, and original work presenting novel research on this recent and important topic.

However, some issues should be addressed, and the English should be reviewed as several typos or grammatic errors are found:

Major comments:

  1. A main concern of this work is the comparison made by the authors between SARS-CoV-2 in humans and the induced one in zebrafish. For instance, there are obvious physiological and biological differences although the sharing similar molecular pathways which are not described in this work. It is referred that SARS-CoV-2 infection alters the innate immune response in human lungs, an organ that is not comparable to the gills of zebrafish. Yet, nothing is included in the manuscript if the responses of SARS-CoV-2 are similar in both organs. Authors should comment on this. In addition, the conclusion of the work should be reviewed taking this into consideration.

Minor comments:

  1. Review the abstract to include further details on the concentrations used, results obtained and conclusions.
  2. Line 251, rather than including microscopic imagens of the swimbladder, which are very subjective, it is suggested to include a quantitative analysis of the area of the swimbladder in each group and the statistics resultant from that analysis. A similar approach could be used for the kidney analysis.
  3. Line 45 is the same info as in line 34.
  4. Review the references style along the manuscript. Ex: [35][36][37][38][39] should be changed to [35-39].
  5. Review the italics for Latin words.
  6. Include the number attributed to the different compounds (table 1) in the chromatogram shown in Figure 1.
  7. Remove references from the results section.
  8. Line 138-193, this information could be fitted in the methodology section as no results is shown.
  9. The information contained in Table 2 is the same as in Figure 3B. Review. Also, the Figure 3a is not a result but an exemplification of the methodology and as such should be moved to the methods section.
  10. Line 235, statistics are missing for table 2. Also, the time spent is shown in what units?
  11. Line 502-505 is equal to line 103-107 but the references are different. Review.
  12. Review the proportion of water and methanol for the extraction. Usually, 80% methanol is used but the contrary is reported.
  13. Line 577, how was zebrafish weight measured? Were the animals sedated to this?
  14. Section 4.11 should be moved to an earlier subtopic.
  15. Topics 4.13, 4.14 and 4.15 should be a subtopic of 4.12.
  16. Line 681, was data checked for normality before applying parametric analysis? Also, parametric data should be presented as mean and SD not SEM while nonparametric data should be described as median and ranges.

Author Response

Major comments:

  1. A main concern of this work is the comparison made by the authors between SARS-CoV-2 in humans and the induced one in zebrafish. For instance, there are obvious physiological and biological differences although the sharing similar molecular pathways which are not described in this work. It is referred that SARS-CoV-2 infection alters the innate immune response in human lungs, an organ that is not comparable to the gills of zebrafish. Yet, nothing is included in the manuscript if the responses of SARS-CoV-2 are similar in both organs. Authors should comment on this. In addition, the conclusion of the work should be reviewed taking this into consideration.

Response: We thank the esteemed reviewer for these keen observations. We agree that there are obvious physiological and biological differences, between zebrafish and humans. We intended to exploit the possible similarities between human and zebrafish and to use zebrafish to model human disease. We have considered the “swim bladder” of zebrafish as physiological relevant air containing organs, like lungs in higher mammals, to xeno-transplant human lung cells in there, and to model SARS-CoV-2 spike protein-induced immune responses.

We have the following updated text in the manuscript (line 74-87) to discuss this a bit more clearly:   

Zebrafish is a small and versatile organism that is very easy to handle and the signaling mechanisms, interaction with chemical modifiers, and the host-virus communications on mucosal tissues are amenable to analysis with delicate detail [17]. Notwithstanding obvious physiological and biological differences, zebrafish has been a model of choice for studying human diseases including virus-induced diseases [18,19], and to discover new drugs or to repurpose existing drugs [20]. Zebrafish has a well-defined innate and adaptive immune systems and shares a remarkable similarity to human counterparts [21]. Further, the interaction between viral glycoproteins and olfactory sensory neurons of the fish nasal region are well understood [22]. Several human viruses including chikungunya, influenza can colonize zebrafish [23] making it a very attractive and alternative model system. Unlike murine models, zebrafish have swim bladders as buoyancy organs, and human cells could be introduced to swim bladders for the generation of xeno-transplanted humanized models for respiratory diseases, like SARS-CoV-2 infection. This approach has been successfully used for modelling lung cancer in zebrafish [24]; and in modelling COPD [25] and P. aeruginosa pathogenesis [26].

Minor comments:

1. Review the abstract to include further details on the concentrations used, results obtained and conclusions.

Response: The abstract has been modified to include the relevant details as suggested by the reviewer.

2. Line 251, rather than including microscopic imagens of the swimbladder, which are very subjective, it is suggested to include a quantitative analysis of the area of the swimbladder in each group and the statistics resultant from that analysis. A similar approach could be used for the kidney analysis.

Response: Thank you very much for this excellent suggestion. We appreciate it very much. However, our present histopathologic assessment software does not have morphometric components for the advised quantitative analysis. We have taken this excellent suggestion and have ordered that specific software module for our next upcoming studies. We are very much thankful for this important insight, which would add one more dimension to our next study plan.

3. Line 45 is the same info as in line 34.

Response: Our apologies for this hindsight. Required changes have been made to address the issue of the repetition of information.

4. Review the references style along the manuscript. Ex: [35][36][37][38][39] should be changed to [35-39].

Response: Thank you, in-text reference citation style has been updated. Yes, it does read and flow better this way.

5. Review the italics for Latin words.

Response: The suggested formatting changes have been made throughout the manuscript.

6. Include the number attributed to the different compounds (table 1) in the chromatogram shown in Figure 1.

Response: Table 1 has been re-formatted based on the peak numbers and retention times for different compounds, as seen in the chromatogram described in Figure 1.

7. Remove references from the results section.

Response: Intext reference citations have been removed from the results. 

8. Line 138-193, this information could be fitted in the methodology section as no results is shown.

Response: We have edited this section considerably and have moved the required text to the method sections, as advised.

9. The information contained in Table 2 is the same as in Figure 3B. Review. Also, the Figure 3a is not a result but an exemplification of the methodology and as such should be moved to the methods section.

Response: Thank you for this keen observation. Erstwhile Figure 3a has now been moved to the methodology section, and have been labelled as new Figure 10. Relevant changes have also been made in the manuscript text and in the Figure legends accordingly.

10. Line 235, statistics are missing for table 2. Also, the time spent is shown in what units?

Response: Our apologies for the hindsight. Statistics have now been added to the table. Time unit (seconds) has also been updated.

11. Line 502-505 is equal to line 103-107 but the references are different. Review.

Response: We thank the esteemed reviewer again. We have now updated the text and changes have been made in accordance with the references cited, and to avoid repetitions.

12. Review the proportion of water and methanol for the extraction. Usually, 80% methanol is used but the contrary is reported.

Response: Our apologies for this typographical error. This has now been corrected in the updated manuscript.

13. Line 577, how was zebrafish weight measured? Were the animals sedated to this?

Response: Individual zebrafish were weighed in a pre-weighed container with a known weight of water, at the desired temperature. This information was included in the methods section and has been clarified further.

14. Section 4.11 should be moved to an earlier subtopic.

Response: Thank you. This section has been moved as advised.

15. Topics 4.13, 4.14 and 4.15 should be a subtopic of 4.12.

Response: The methods for in-vitro experiments have been now been combined together in one topic, and various assays have been described as sub-headings.

16. Line 681, was data checked for normality before applying parametric analysis? Also, parametric data should be presented as mean and SD not SEM while nonparametric data should be described as median and ranges.

Response: Yes, the data were checked for normality. We have also checked for SD vs. SEM presentation of data. We have found both ways have reported in various concurrent literature (for example: SEM used along with  ANOVA in Audira et al., Biomolecules 2020, 10. Doi: 10.3390/biom10091340). To make data descriptions clearer, we have now updated the ‘n’ values in all the figure legends along with SEM description.

Thank you very much indeed, for these elaborate comments and suggestions. We feel these suggested changes have made our manuscript much better and added more clarity to study outcomes. We appreciate your kindness and efforts for these valuable suggestions.  

Round 2

Reviewer 3 Report

The authors made all changes suggested. The manuscript can be accepted.

Reviewer 4 Report

The authors have made substantial changes to the manuscript, greatly improving its quality and readibility addressing all my comments. Therefore, I have no more comments to the authors.